# HyperDAS: Towards Automating Mechanistic Interpretability with Hypernetworks

**Jiuding Sun**[♡]          **Jing Huang**[♦]          **Sidharth Baskaran**[♦]          **Karel D'Oosterlinck**[♣]

**Christopher Potts**[♦]                    **Michael Sklar**[*♦]                    **Atticus Geiger**[*♡]

[♡]**Pr(Ai)$^2$R Group**      [♦] **Stanford University**      [♦] **Confirm Labs**      [♣] **Ghent University**

## Abstract

Mechanistic interpretability has made great strides in identifying neural network features (e.g., directions in hidden activation space) that mediate concepts (e.g., *the birth year of a person*) and enable predictable manipulation. Distributed alignment search (DAS) leverages supervision from counterfactual data to learn concept features within hidden states, but DAS assumes we can afford to conduct a brute force search over potential feature locations. To address this, we present HyperDAS, a transformer-based hypernetwork architecture that (1) automatically locates the token-positions of the residual stream that a concept is realized in and (2) constructs features of those residual stream vectors for the concept. In experiments with Llama3-8B, HyperDAS achieves state-of-the-art performance on the RAVEL benchmark for disentangling concepts in hidden states. In addition, we review the design decisions we made to mitigate the concern that HyperDAS (like all powerful interpretabilty methods) might inject new information into the target model rather than faithfully interpreting it.

## 1 Introduction

Mechanistic interpretability methods promise to demystify the internal workings of black-box language models (LMs), thereby helping us to more accurately control these models and predict how they will behave. Automating such efforts is critical for interpreting our largest and most performant models, and strides toward this goal have been made for circuit discovery (Conmy et al., 2023; Rajaram et al., 2024) and neuron / feature labeling (Bills et al., 2023; Huang et al., 2023; Schwettmann et al., 2023; Shaham et al., 2024). In the present paper, we complement these efforts by taking the first steps toward automating interpretability for identifying features of hidden representations (e.g., directions in activation space) that mediate concepts (Mueller et al., 2024; Geiger et al., 2024a).

Interventions on model-internal states are the building blocks of mechanistic interpretability (Saphra & Wiegreffe, 2024). To establish that features of a hidden representation are mediators of a concept, a large number of *interchange intervention* (Vig et al., 2020; Geiger et al., 2020) experiments are performed on the LM. Interchange interventions change features to values they would take on if a counterfactual input were processed. For example, if the concept is $C$ = *the birth year of a person*, we can fix the features $F$ of an LM processing the input *Albert Einstein was born in* to the value they take for *Marie Curie was a chemist*. If the output changes from *1879* to *1934*, we have a piece of evidence that $F$ mediates $C$. The field has developed a variety of methods for learning such interventions, but all of them require a brute-force search through potential hidden representations.

To address this significant bottlebeck, we propose HyperDAS, a method to automate this search process via a hypernetwork. In the HyperDAS architecture, a transformer-based hypernetwork localizes a concept within the residual stream of a fixed layer in a target LM by:

1. Encoding a language description (e.g., *the birth year of a person*) of a concept using a transformer that can attend to the target LM processing a *base prompt* (e.g., *Albert Einstein was born in*) and a *counterfactual prompt* (e.g., *Marie Curie was a chemist*).

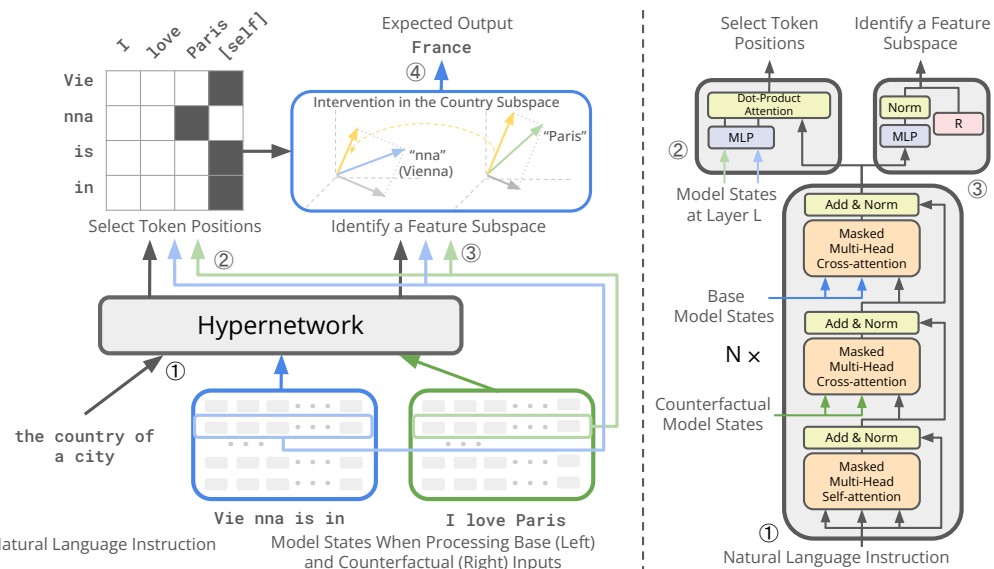

Figure 1: **The HyperDAS framework**, used here to find the features that mediate the concept of "country". **(1) Concept Encoding** A natural language description that specifies which concept to localize, "The country of a city", is encoded by a transformer hypernetwork with two additional cross-attention blocks attending to the hidden states of the target LM prompted with the base text "Vienna is in" and the counterfactual text "I love Paris". **(2) Selecting Token-Positions** With the encoding from step 1 as a query, HyperDAS uses selects the tokens "nna" and "Paris" as the location of the concept "country" for the base and counterfactual, respectively. **(3) Identifying a Subspace** With the representation from step 1 as the encoding, HyperDAS constructs a matrix whose orthogonal columns are the features for "country". **(4) Interchange Intervention** With the token-positions from step 2 and subspace from step 3, HyperDAS performs a intervention by patching the subspace of the hidden vector for the token "nna" to the value it takes on in the hidden vector for the token "Paris", leading the model to predict "France" from the base prompt "Vienna is in".

2. Pairing tokens in the base and counterfactual prompts (e.g., align "Cur" with "Ein") with an attention mechanism using the encoding from (1) as a query and token-pairs as keys/values.

3. Selecting features of the residual stream via a fixed orthogonal matrix that undergoes a Householder transformation (Householder, 1958) using the encoding from (1).

4. Patching the selected residual stream features of aligned tokens from the base prompt to the values they take on in the residual stream of aligned tokens from the counterfactual prompt.

We benchmark HyperDAS on the RAVEL interpretability benchmark (Huang et al., 2024), in which concepts related to a type of entity are disentangled. For example, we might seek to separate features for the *birth year* and *field of study* of a Nobel laureate. HyperDAS achieves state-of-the-art performance on RAVEL with a single model. Greater gains are achieved when a separate HyperDAS is trained for each entity type (e.g., *Nobel laureates*).

Finally, we address the issue of whether HyperDAS is faithful to the target model. As we use more complex machine learning tools for interpretability, there is an increasing concern that we are not uncovering latent causal structure, but instead injecting new information to steer or edit a model (Meng et al., 2022; Ghandeharioun et al., 2024). If we allow our supervised interpretability models too much power, we run the risk of false-positive signals. Thus, we conclude with a discussion of how our decisions about architecture, training, and evaluation were made to mitigate these concerns.

## 2 BACKGROUND

**Automating Interpretability Workflows** The growing size and complexity of language models demands scalable techniques for interpretability. Major directions include circuit analysis (Conmy

et al., 2023; Marks et al., 2024; Rajaram et al., 2024; Ferrando & Voita, 2024), unsupervised feature learning (Huben et al., 2024; Braun et al., 2024), and feature labeling with natural language descriptions (Mu & Andreas, 2021; Hernandez et al., 2022; Bills et al., 2023; Huang et al., 2023; Shaham et al., 2024). In this work, we take steps towards automating the process of identifying features that are causal mediators of concepts using supervision from counterfactual data.

**Identifying Features that Mediate Concepts**    Interchange interventions identify neural representations that are causal mediators of high-level concepts (Vig et al., 2020; Geiger et al., 2020; Finlayson et al., 2021; Stolfo et al., 2023). Geiger et al. (2024b) and Wu et al. (2024) further extend interchange interventions to localizing concepts in hidden vector subspaces. However, these methods require an exhaustive search over all layers and tokens to measure causal effects at each position. In practice, the lack of an effective search method leads to heuristics in token selection. For example, in knowledge editing and model inspection, a widely held assumption is that the entity information is localized to the last entity token (Meng et al., 2022; 2023; Hernandez et al., 2024; Geva et al., 2023; Ghandeharioun et al., 2024), but this does not hold for all entities (Meng et al., 2022). Our proposed method directly addresses this problem by using an end-to-end optimization to automatically select the intervention site across all tokens, conditioned on the concept to localize.

**The RAVEL Benchmark**    The RAVEL benchmark evaluates how well an interpretability method can localize and disentangle entity attributes through causal interventions. An example consists of a base prompt that queries a specific attribute of an entity (e.g., _Albert Einstein studied the field_), a counterfactual prompt containing a different entity of the same type (e.g., _Poland declared 2011 the Year of Marie Curie_), an attribute targeted for intervention (e.g., _field of study_ or _birth year_), and a counterfactual label for the base prompt. The label would be _physics_ if the targeted attribute is _birth year_ (the intervention should not affect _the field of study_ attribute), and it would be _chemistry_ if the targeted attribute is _field of study_.

**Distributed Interchange Interventions**    RAVEL supports evaluations with _distributed interchange interventions_ on features of a hidden representation $\mathbf{H}$ that encode an attribute in the original model $\mathcal{M}$. In our experiments, features are orthogonal directions that form the columns of a low-rank matrix $\mathbf{R}$. Given a base prompt $\bar{\mathbf{x}}$ and a counterfactual prompt $\hat{\mathbf{x}}$, we perform an intervention that fixes the linear subspace spanned by $\mathbf{R}$:

$$\mathbf{H} \leftarrow \bar{\mathbf{h}} + \mathbf{R}^\top\big(\mathbf{R}(\hat{\mathbf{h}}) - \mathbf{R}(\bar{\mathbf{h}})\big) \tag{1}$$

where $\bar{\mathbf{h}}$ and $\hat{\mathbf{h}}$ are the values that variable $\mathbf{H}$ has when the model $\mathcal{M}$ is run on $\bar{\mathbf{x}}$ and $\hat{\mathbf{x}}$, respectively.

**RAVEL Metrics**    The metric from the RAVEL dataset has two components. The Cause score is the proportion of interchange interventions that successfully change the attribute that was targeted, and the Iso score is the proportion of interchange interventions that successfully do not change an attribute that was not targeted. The Disentangle score is the average of these two.

**Distributed Alignment Search**    The RAVEL evaluations use distributed alignment search (DAS; Geiger et al. 2024b) as a baseline for learning a feature of a hidden representation that mediate an attribute. A rotation matrix is optimized on a RAVEL example with base input $\bar{\mathbf{x}}$, counterfactual input $\hat{\mathbf{x}}$, and counterfactual label $y$ using the following loss:

$$\mathcal{L}_{\mathsf{DAS}} = \mathsf{CE}\big(\mathcal{M}_{\mathbf{H} \leftarrow \bar{\mathbf{h}} + \mathbf{R}^\top\big(\mathbf{R}(\hat{\mathbf{h}}) - \mathbf{R}(\bar{\mathbf{h}})\big)}(\bar{\mathbf{x}}), y\big) \tag{2}$$

where $\mathcal{M}_\gamma(\bar{\mathbf{x}})$ is the output of the model $\mathcal{M}$ run on input $\bar{\mathbf{x}}$ with an intervention $\gamma$. Only the parameters $\mathbf{R}$ are updated while the parameters of the target model $\mathcal{M}$ are frozen.

## 3    HYPERDAS

To localize a concept in the layer $l$ of a target model $\mathcal{M}$, a HyperDAS architecture consists of a hypernetwork $\mathcal{H}$ that takes in a text specification $\mathbf{x}$ of the target concept and dynamically selects token positions in the base text $\bar{\mathbf{x}}$ and counterfactual text $\hat{\mathbf{x}}$ and identifies a linear subspace $\mathbf{R}$ that mediates the target concept. Selecting tokens is a discrete operation, so we "soften" the selection during training and force discrete decisions during evaluation. Our specific model is as follows.

## 3.1 REPRESENTING THE TARGET CONCEPT AS A VECTOR

**Token Embedding** A token sequence $\mathbf{x}$ of length $E$ that specifies the concept to localize, e.g., *the country a city is in*, is encoded with the embeddings of the target model $\mathcal{M}$ to form $\mathbf{e}_0 = \text{Emb}(\mathbf{x}) \in \mathbb{R}^{E \times d}$, the zeroth layer of the residual stream for the hypernetwork $\mathcal{H}$.

**Cross-attention Decoder Layers** After embedding the target concept, we run a transformer with $N$ decoder layers. Besides the standard multi-headed self-attention (MHA) and feed-forward layers (MLP), each decoder block has two additional cross-attention modules to incorporate information from the target model $\mathcal{M}$ procesing the base and counterfactual prompts $\bar{\mathbf{x}}$ and $\hat{\mathbf{x}}$.

Let $\bar{\mathbf{h}} \in \mathbb{R}^{B \times L \times d}$ and $\hat{\mathbf{h}} \in \mathbb{R}^{C \times L \times d}$ be the stacks of base and counterfactual hidden states from the base and the counterfactual input, where $L$ is the number of layers in $\mathcal{M}$, $d$ is the hidden dimension, and $B$ and $C$ are the sequence length of the base and counterfactual examples, respectively. Two multi-headed cross-attention modules $\text{M}\bar{\text{H}}\text{A}$ and $\text{M}\hat{\text{H}}\text{A}$ allow $\mathcal{H}$ to attend to $\bar{\mathbf{h}}$ and $\hat{\mathbf{h}}$. Each layer of the hypernetwork $\mathcal{H}$ can attend to every layer of the target model.

For the $p$-th decoder layer of the hypernetwork $\mathcal{H}$, the three attention mechanisms are as follows:

$$\mathbf{e}_p' = \text{MHA}\big(\mathbf{Q}(\mathbf{e}_p), \mathbf{K}(\mathbf{e}_p), \mathbf{V}(\mathbf{e}_p)\big) \tag{3}$$

$$\mathbf{e}_p'' = \text{M}\bar{\text{H}}\text{A}\big(\bar{\mathbf{Q}}(\mathbf{e}_p'), \bar{\mathbf{K}}(\bar{\mathbf{h}}), \bar{\mathbf{V}}(\bar{\mathbf{h}})\big) \tag{4}$$

$$\mathbf{e}_{p+1} = \text{M}\hat{\text{H}}\text{A}\big(\hat{\mathbf{Q}}(\mathbf{e}_p''), \hat{\mathbf{K}}(\hat{\mathbf{h}}), \hat{\mathbf{V}}(\hat{\mathbf{h}})\big) \tag{5}$$

After the final transformer block is applied, the residual stream vector at the last token position $\mathbf{e}_E^{(N)} \in \mathbb{R}^d$ encodes information about the concept targeted for intervention and the target model's base and counterfactual runs. This representation is used to select token-positions in the base and source texts and identify a linear subspace for intervention.

## 3.2 DYNAMICALLY SELECTING TOKEN-POSITIONS THAT CONTAIN THE TARGET CONCEPT

The next step is selecting the tokens in the base and counterfactual prompts that contain the concept encoded as $\mathbf{e}_E^{(N)}$. We construct an "intervention score" matrix $G \in \mathbb{R}^{B \times (C+1)}$ with elements ranging from 0 to 1, where $B$ and $C$ are the number of tokens in the base and counterfactual prompts, respectively. Each element $G_{(b,c)}$ denotes the degree to which the $b$-th base token is aligned with $c$-th counterfactual token for intervention. The additional column $G_{(b,c)}$ corresponds to the score for retaining the $b$-th base token without any intervention.

In Figure 1, only the base-token "nna" (the last token of the entity "Vienna") receives an intervention score of 1 when paired with the counterfactual=token "Paris". All other base tokens are not selected for intervention. The concept of "country" is localized to the last token of the city entities.

To compute the element $G_{(b,c)}$, we use $\bar{\mathbf{h}}_b^{(l)}$ and $\hat{\mathbf{h}}_c^{(l)}$, the $l$-th layer residual stream representation of the target model at $b$-th base-token and $c$-th counterfactual-token. We first linearly combine the two:

$$\mathbf{g}_{(b,c)} = F([\bar{\mathbf{h}}_b^{(l)}; \hat{\mathbf{h}}_c^{(l)}]) \tag{6}$$

where $F(.) : \mathbb{R}^{2d} \to \mathbb{R}^d$ is a linear projection that condenses the concatenated representation into the original dimension $d$. For the extra column that indicates no intervention, the representation is simply the original base token representation:

$$\mathbf{g}_{(b,C+1)} = \bar{\mathbf{h}}_b^{(l)} \tag{7}$$

Then, we weight each merged representation $\tilde{\mathbf{h}}$ using the concept encoding $\mathbf{e}_E^{(N)} \in \mathbb{R}^d$ and $M$ "query" and "key" matrices $Q^{(i)} \in \mathbb{R}^{d \times M \times \frac{d}{M}}$ and $K^{(i)} \in \mathbb{R}^{d \times M \times \frac{d}{M}}$:

$$G_{(b,c)}' = \frac{\sum_{i=1}^{M} \big(\mathbf{e}_E^{(N)}\big) Q^{(i)} \cdot \big(\tilde{\mathbf{h}}\big) K^{(i)}}{M\sqrt{d}} \tag{8}$$

Finally, we apply a column-wise softmax $G = \text{ColumnSoftmax}(G')$.

Using the matrix $G$, we can construct the representation that we will use to intervene on each token in the base prompt. For the $b$-th base-token hidden states $\bar{\mathbf{h}}_b^l$, the interventio representation is:

$$\tilde{\mathbf{h}}_b^{(l)} = G_{(b,C+1)}\bar{\mathbf{h}}_b^{(l)} + \sum_{c=1}^{C} G_{(b,c)}\hat{\mathbf{h}}_c^{(l)} \tag{9}$$

This is how the counterfactual representation is constructed for a weighted interchange intervention (Wu et al., 2024). The counterfactual hidden states remain identical to the base hidden states (i.e., no intervention on token $b$) when $G_{(b,C+1)} = 1$. Conversely, if $G_{(b,c)} = 1$ for a specific position $c$, the $b$-th base token is entirely replaced by the hidden vector for the $c$-th counterfactual token.

## 3.3 Dynamically Identifying a Linear Subspace that Contains the Concept

In addition to selecting token-positions, HyperDAS also dynamically identifies a linear subspace that contains the target concept, encoded as a low-rank matrix with orthogonal columns. First, we apply a multi-layer perceptron to $\mathbf{e}_E^{(N)} \in \mathbb{R}^d$ in order to produce a new vector

$$\mathbf{v} = \mathsf{MLP}(\mathbf{e}_E^{(N)}) \in \mathbb{R}^d.$$

In DAS, there is a fixed low-rank matrix with orthogonal columns $\mathbf{R}'$ representing a fixed subspace targeted for intervention. We use a linear algebra operation known as the Householder transformation to change $\mathbf{R}'$ conditional on $\mathbf{v}$ into a new matrix $\mathbf{R}$ that still has orthogonal columns. Given a non-zero vector $\mathbf{v} \in \mathbb{R}^d$, the Householder transformation $\mathbf{H}$ is defined as:

$$\mathbf{H} = \mathbf{I} - 2\frac{\mathbf{v}\mathbf{v}^\top}{\mathbf{v}^\top\mathbf{v}}, \tag{10}$$

where $\mathbf{I}$ is the identity matrix. The matrix $\mathbf{H}$ is orthogonal and $\mathbf{R}'$ has orthogonal columns, which means $\mathbf{R}'\mathbf{H}$ has orthogonal columns. Utilizing this property, we can dynamically select the subspace based on the intervention representation $\mathbf{e}_E^{(N)}$ by computing $\mathbf{R} = \mathbf{R}'\mathbf{H}$.

## 3.4 Intervening on the Subspace at the Selected Token-Positions

After selecting token-positions and identifying a subspace, we finally intervene. For each base-token $b$ in the $l$-th layer of the target model $\mathcal{M}$, we perform a weighted interchange intervention with the counterfactual hidden states $\tilde{\mathbf{h}}^{(l)}$ and low-rank orthogonal matrix $\mathbf{R}$:

$$\mathbf{H}_b^{(l)} \leftarrow \bar{\mathbf{h}}_b^{(l)} + \mathbf{R}^\top\big(\mathbf{R}(\tilde{\mathbf{h}}_b^{(l)}) - \mathbf{R}(\bar{\mathbf{h}}_b^{(l)})\big). \tag{11}$$

## 3.5 Training

We train HyperDAS on the RAVEL with two losses. The first simply measures success on RAVEL. The second incentivizes the model to select unique token-pairings so that performance is maintained when the token alignment matrix $G$ is snapped to binary values during test-time evaluation.

**RAVEL Loss**  A RAVEL example consists of a base input $\bar{\mathbf{x}}$, counterfactual input $\hat{\mathbf{x}}$, target concept input $\mathbf{x}$, and a counterfactual label $\mathbf{y}$. When the target concept matches the attribute queried in the base input, the label $\mathbf{y}$ is the attribute of $\hat{\mathbf{x}}$. Otherwise, $\mathbf{y}$ is the attribute of $\bar{\mathbf{x}}$. The loss is:

$$\mathcal{L}_{\text{RAVEL}} = \text{CE}\big(\mathcal{M}_{\mathbf{H}_b^{(l)} \leftarrow \bar{\mathbf{h}}_b^{(l)} + \mathbf{R}^T\big(\mathbf{R}(\tilde{\mathbf{h}}_b^{(l)}) - \mathbf{R}(\bar{\mathbf{h}}_b^{(l)})\big)}(\bar{\mathbf{x}}), \mathbf{y}\big) \tag{12}$$

**Sparse Attention Loss**  The construction of $G$ allows for a single counterfactual token to be paired with a weighted combination of multiple base tokens, however during test-time evaluations we will enforce a 1-1 correspondence. Thus, we include a sparse attention loss that penalizes cases where one counterfactual token attends strongly to multiple base tokens in each row of matrix $G$:

$$\mathcal{L}_{\text{sparse}} = \frac{1}{C}\sum_{c=1}^{C}\begin{cases}\text{Sum}(G_{(*,c)}) & \text{if } \text{Sum}(G_{(*,c)}) > 1 \\ 0 & \text{if } \text{Sum}(G_{(*,c)}) \leq 1\end{cases} \tag{13}$$

The final loss is $\mathcal{L} = \mathcal{L}_{\text{RAVEL}} + \lambda\mathcal{L}_{\text{sparse}}$, where $\lambda$ is a real-valued weight scheduled during training.

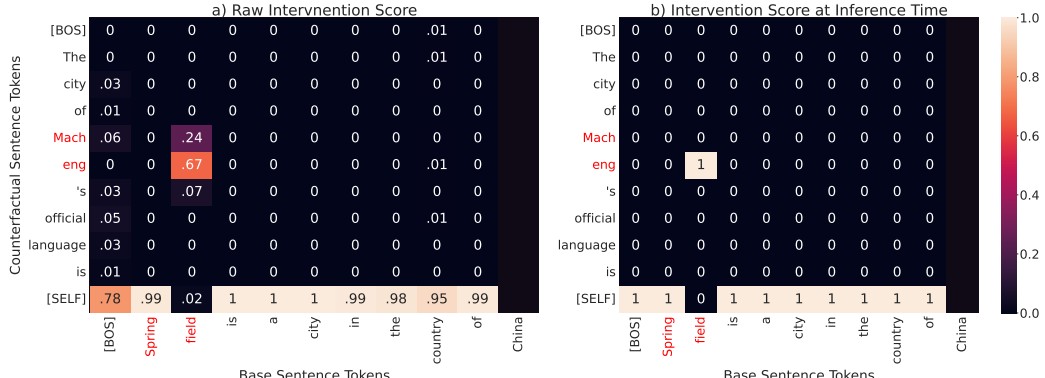

Figure 2: The "intervention score" matrix $G$ for the counterfactual prompt *"The city of Macheng's official language is"* and base prompt *"Springfield is a city in the country of"*, for which the target model will output *"China"*. The attribute targeted for intervention is *country*, so the output should be *"The United States"*. The raw intervention (left) is produced by the token-position selection discussed in Sec 3.2, and a column-wise and row-wise argmax is applied at inference time to enforce an 1-1 correspondence between the base and counterfactual tokens, detailed in Sec. 3.6.

## 3.6 EVALUATION

HyperDAS is end-to-end differentiable because discrete operations like aligning base and counterfactual tokens are "softened" using softmax operators. During the evaluation, we force these discrete decisions. In many cases, the matrix $G$ contains non-zero weight for multiple base-counterfactual sentence token pairs (**left** panel in Figure 2). At inference time, we perform a double argmax operation on the intervention score to select the most important location for the intervention. For each base-counterfactual token pair, it is set to 1 if and only if this position gets the highest intervention score across its row and column.

$$
G_{(b,c)} = \begin{cases} 1 & \text{if } G_{(b,c)} = \max(G_{(b,*)}, G_{(*,c)}) \\ 0 & \text{otherwise} \end{cases} \tag{14}
$$

The [SELF] row, representing no intervention on a base-token, is readjusted according to the discrete intervention weight (**right** panel in Figure 2).

$$
G_{(b,C+1)} = 1 \text{ if } \max(G_{(b,*)}) = 0 \tag{15}
$$

## 4 EXPERIMENTS

We benchmark HyperDAS on RAVEL with Llama3-8B (Meta, 2024) as the target model. We both train a HyperDAS model on all of RAVEL at once and also train a separate HyperDAS model for each of the five entity domains in the RAVEL benchmark, i.e., *cities*, *Nobel laureates*, *occupations*, *physical objects*, and *verbs*. We experimented with initializing the transformer hypernetwork from pre-trained parameters, but found no advantage for this task.

**Crucial Hyperparameters**   We use 8 decoder blocks for the hypernetwork and 32 attention heads for computing the pairwise token position attention. The sparsity loss weight is scheduled to linearly increase from 0 to 1.5, starting at 50% of the total steps. A learning rate between $2 \times 10^4$ to $2 \times 10^5$ is chosen depending on the dataset. Discussion of these choices concerning the sparsity loss is in Section 4.2. For the feature subspace, we experiment with dimensions from 32 up to 2048 (out of 4096 dimensions) and use a subspace of dimension 128.

**Masking of the Base Prompt**   As the hypernetwork has access to the target attribute information from the instruction and the base attribute information from the base model states, a trivial solution

| Methods | City | | Nobel Laureate | | Occupation | | Physical Object | | Verb | | Average |
|---|---|---|---|---|---|---|---|---|---|---|---|
| | Causal | Iso | Causal | Iso | Causal | Iso | Causal | Iso | Causal | Iso | Disentangle |
| MDAS | 55.8 | 77.9 | 56.0 | 93.5 | 50.7 | 88.1 | 85.0 | 97.9 | 74.3 | 79.6 | 76.0 |
| HyperDAS | | | | | | | | | | | |
| - Asymmetric | 70.8 | 93.9 | 55.4 | 95.1 | 50.4 | 99.1 | 92.7 | 97.2 | 93.0 | 98.9 | 84.7 |
| - Asymmetric All Domains | 58.8 | 90.5 | 47.6 | 92.0 | 75.7 | 82.1 | 92.9 | 94.5 | 86.9 | 95.8 | 80.7 |
| - Symmetric | 76.9 | 90.9 | 59.2 | 88.4 | 47.1 | 89.1 | 94.8 | 97.5 | 42.3 | 82.9 | 76.9 |
| - Symmetric All Domains | 16.8 | 94.7 | 2.0 | 98.8 | 6.1 | 97.3 | 21.6 | 99.3 | 13.6 | 97.8 | 54.8 |

(a) Main results of HyperDAS on five domains of RAVEL with Llama3-8B. For each method, we report the results from the best layer between 10 and 15. HyperDAS achieves state-of-the-art attribute disentangling performance across the board. The MDAS baseline intervenes on fixed tokens.

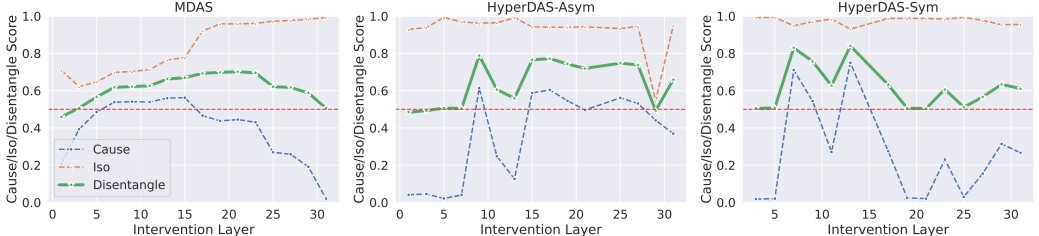

(b) The cause/iso/disentangle score of the baseline method and HyperDAS (Both symmetric and asymmetric implementation) for the entity type of "city" across the layers of Llama3-8B. For both HyperDAS and MDAS, the peak Disentangle Score happens around L15.

Figure 3: RAVEL benchmark results. HyperDAS establishes a new state-of-the-art.

the hypernetwork can learn is to condition the intervention location on whether the target attribute matches the base attribute. In other words, if the two attributes match, attend to a location that has causal effect on the output, otherwise attend to the extra [self] row (see Appendix A.3 for an example). This solution, however, does not find the actual concept subspace. To prevent the hypernetwork from learning this trivial solution, we apply an attention mask to the base prompt to mask out the attribute information. With the masking, the hypernetwork no longer has access to the base attribute information, and hence the localization prediction is only conditioned on the target attribute in the natural language instruction.

**Symmetry** Intuitively, if we have localized a concept, then "get" operations that retrieve the concept and "set" operations that fix the concept should both target the same features and hidden representations. For this reason, we consider a variant of HyperDAS that enforces symmetry in the localization of base and counterfactual prompts. We can enforce symmetry between base and counterfactual inputs during token selection by randomly flipping the order of the concatenation between base and counterfactual hidden representations in Equation 6. We report full results for symmetric and asymmetric models.

**Multi-task DAS (MDAS) Baseline** The current state-of-the-art method on RAVEL is MDAS, which uses a multi-task learning objective to satisfy multiple high-level causal criteria. MDAS requires supervised training data like HyperDAS, however, MDAS relies on manually selected token position for intervention, which in our case is the final token of the entity, e.g., "nna" in Figure 1.

**Results** In Table 3a, we show results on RAVEL for layer 15 of Llama3-8B. In Figure 3b, we also run HyperDAS targeting every 2 layers in Llama3-8B starting from the embedding layer. The peak performance of attribute disentanglement for both MDAS and HyperDAS is around layer 15.

## 4.1 LAYER-SPECIFIC INTERVENTION BEHAVIORS OF HYPERDAS

HyperDAS searches for an optimal location to intervene within the target hidden state in a chosen layer. We evaluate MDAS and HyperDAS on 16 layers across the model (Figure 3b) and chose an early layer, middle layer, and deep layer for detailed study: Layer 7, Layer 15, and Layer 29.

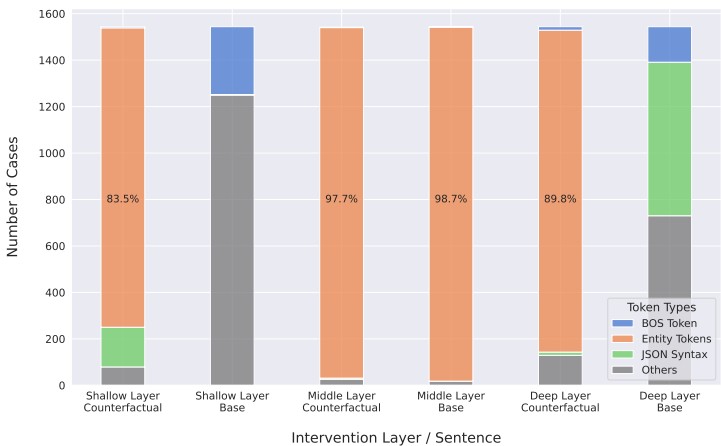

Figure 4: The intervention location, in counterfactual and base sentence, picked by HyperDAS when targeting shallow (7), middle (15) and deep (29) decoder layers.

Analysis presented in Figure 4 reveals that HyperDAS consistently targets entity tokens in the counterfactual input across all layers, suggesting robust detection of attribute information in the entity token's residual stream from an early stage. However, the choice of intervention location within the base input shows significant variation. For each example in the "city" entity split, we categorize the base and counterfactual token pair that gets the *largest* intervention weight, and classify them into the following categories: (1) **BOS Token** represents the beginning-of-sentence token. (2) **Entity Token** refers to tokens representing entities. (3) **JSON Syntax** includes special characters and syntactic tokens typical of JSON formatted text (e.g., opening curly brace "{"). (4) **Others** comprises all tokens irrelevant to the current analysis, with "is" following the entity token being a common example in both shallow (36%) and deep (29%) layer bases.

At very early layers, HyperDAS displays turbulent behavior, targeting random or even beginning-of-sentence tokens in the base sentence. By the middle layers, the model consistently favors the entity token for intervention, aligning with findings from Huang et al. (2024) and Geva et al. (2023). In contrast, at deeper layers, the hypernetwork learns to intervene on unintuitive positions such as syntax tokens within a JSON-formatted prompt, which were previously unknown to store attributes.

## 4.2 DISCUSSION

**HyperDAS establishes a new state-of-the-art performance on RAVEL**   Our results show that HyperDAS outperforms MDAS, the previous state-of-the-art, across all entity splits at layer 15 in Llama3-8B and across all layers of Llama3-8B for the "cities" entity split.

**HyperDAS requires more compute than MDAS.**   HyperDAS is more powerful than MDAS, but also more computationally expensive. Training our HyperDAS model for one epoch on disentangling the country attribute in the city domain takes 468,923 TeraFLOPs, while training an MDAS model for one epoch on the same task takes 193,833. Thus, HyperDAS requires ≈ 2.4x compute.

On the other hand, HyperDAS is more memory efficient for tasks like RAVEL. Our target Llama model requires 16GB of RAM. The HyperDAS model requires 52GB more in total for RAVEL, whereas MDAS requires only 4.1GB more per attribute. The memory usage of HyperDAS does not go up with additional attributes, so when trained on all of RAVEL together (23 attributes), MDAS (23 * 4.1 + 16 = 110.3GB) exceeds the memory usage of HyperDAS (52 + 16 = 68GB).

**Householder vectors analysis provides a window into attribute features.**   To analyze the Householder vectors generated by the model, we collected vectors from each test example and categorized them according to their respective attributes. For each attribute category, a subset of 1,000 samples was randomly selected. We then computed the similarity scores between pairs of attributes by calculating the average cosine similarity across these 1,000 pairs of selected Householder vectors.

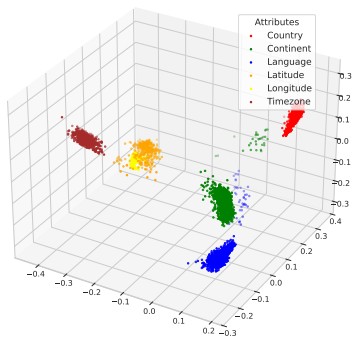

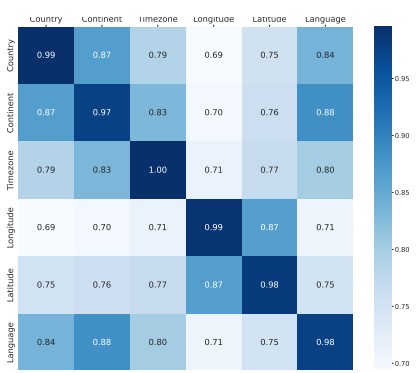

Figure 5: The relative position between the Householder vector (after PCA) of attributes for all the correct predictions in city domain. The clustering indicates that HyperDAS learns different subspace for each attribute.

Figure 6: The cosine similarity between the Householder vectors of different attributes in the city domain, computed using 100,000 samples from each attribute. Notably, HyperDAS effectively learns a highly similar subspace for the attributes 'Longitude' and 'Latitude'.

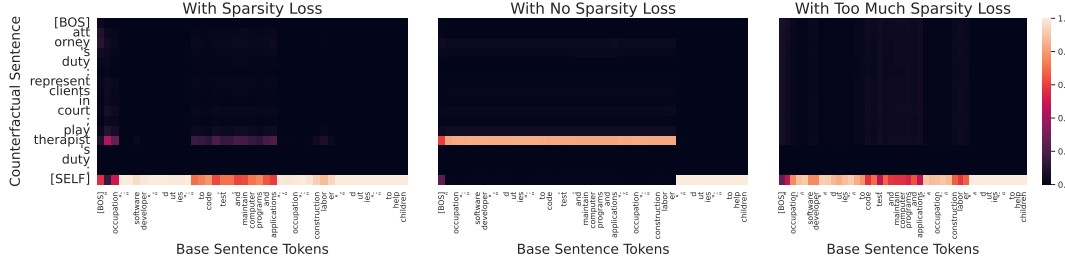

Figure 7: Intervention locations for a base/counterfactual sentences pair with *Occupation* entity-type selected by HyperDAS trained with different amounts of sparsity loss. This comparison illustrates the intervention locations generated by HyperDAS when trained under three different sparsity loss conditions. All three models achieved a Disentangle Score ≈ **94.0**% using weighted interventions. With no sparsity loss (middle), HyperDAS tends to intervene from the last subject token in the counterfactual sentence to most tokens in the base sentence, which yields adequate performance under many-to-one constraints but not under strict one-to-one constraints. With too much sparsity loss (right), the pairwise token selection attention within HyperDAS fails, resulting in interventions that blend all hidden states. Although this approach achieves a near-perfect disentangle score with weighted intervention, the model's does not have interpretable intervention patterns and fails entirely during test time when masks are snapped to align base and source tokens one-to-one.

We analyze the geometry of the learned householder vectors, with the PCA projection shown in Figure 5. We also compute the average pairwise cosine similarity of Householder vectors sampled from within the same attribute or cross two different attributes, as shown in Figure 6. Despite an overall high cosine similarity among all Householder vectors associated with the same entity type, the Householder vectors associated with the same attribute form a tighter cluster, with a higher cosine similarity score than pairs of vectors associated with two different attributes. These per-attribute clusters might explain why the learned subspace can disentangle different attributes of the same entity, as different attributes are localized into different subspaces of the entity representation.

**How do we know HyperDAS uncovers actual causal structures faithful to the target model?**
On one hand, we should leverage the power of supervised machine learning to develop increasingly sophisticated interpretability methods. On the other hand, such methods are incentivized to "hack" evaluations without uncovering actual causal structure in the target model. We have taken several steps to maintain fidelity to the underlying model structures when training and evaluating Hyper-DAS, by constraining optimization flexibility to prevent inadvertently steering or editing the model with out-of-distribution interventions.

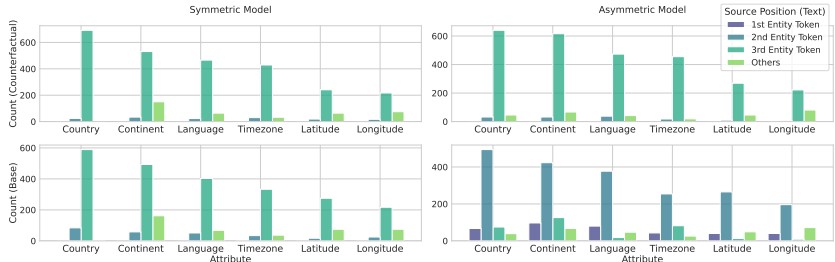

Figure 8: The count of intervention location picked by HyperDAS at the counterfactual prompt (upper) v.s. at the base prompt (bottom) across all the attributes in the city domain on entities with three tokens. The asymmetric variant (right) of HyperDAS favors getting the attribute information from the **last entity token** for the majority of the counterfactual prompts (≥ 95%), and intervene on the **second last entity token**. The symmetric variant (left) favors **last entity token** consistently for both base and counterfactual prompt.

**The weighted interchange interventions used in training hacks the objective without soft constraints via loss terms.** The loss term $\mathcal{L}_{\text{sparse}}$ is crucial for ensuring that HyperDAS learns a one-to-one alignment between base tokens and counterfactual tokens (Figure 2). When no sparsity loss is applied, the model aligns the final entity token (e.g., "nna" from Figure 1) to many tokens in the base sentence. These solutions fail during evaluations where token alignments are snapped to be one-to-one. Conversely, with excessive sparsity loss, the model constructs a counterfactual hidden representation that is a linear combination of many hidden states, resulting in a high flexibility optimization scheme that is closer to model steering or editing. This also fails during one-to-one evaluations. See Figure 7 for an example of these pathological settings.

**Often only one token is aligned between base and counterfactual inputs.** The MDAS baseline performs well on the RAVEL benchmark by one token in the base and one token in the source. However, our new state-of-the-art HyperDAS model will select multiple tokens 53% of the time.

**Asymmetric HyperDAS targets different tokens for base and counterfactual examples.** Figure 8 shows the tokens selected by the symmetric and asymmetric variants of HyperDAS. When allowed asymmetric parametrization, networks break symmetry in positional assignments; for a single input prompt, HyperDAS will select different tokens depending on whether that input is the base or counterfactual.

## 5 CONCLUSION

In this work, we introduced HyperDAS, a novel hypernetwork-based approach for automating causal interpretability work. HyperDAS achieves state-of-the-art performance on the RAVEL benchmark, demonstrating its effectiveness in localizing and disentangling entity attributes through causal interventions. Our method's ability to dynamically select hidden representations and learn linear features that mediate target concepts represents a significant advancement in interpretability techniques for language models. We are optimistic that HyperDAS will open new avenues for understanding and interpreting the internal workings of complex language models.

**Limitations** HyperDAS will only be successful if the target concept is mediated by linear features. However, there is emerging evidence that non-linear mediators are a possibility (Csordás et al., 2024; Engels et al., 2024). As discussed extensively in the main text, applying supervised machine learning to interpretability has the potential to lead to false positive results. While we have taken steps to maintain fidelity to underlying model structures, future work should continue to explore the delicate balance between uncovering latent causal relationships and the risk of model steering.

## ACKNOWLEDGEMENTS

This research was in part supported by a grant from Open Philanthropy.

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

## A APPENDIX

### A.1 HYPERDAS OVER ALL DOMAINS

Our results at Table 3a show that we can train HyperDAS to achieve state of the art performance on the RAVEL benchmark by training a separate model for each entity type split, which is the set up used to train the previous state of the art MDAS. To test the scalibility and generalizability of HyperDAS, we train a single model across all the entity type splits and evaluate its performance.

**Experiment Set-up** We aggregate the training split of the dataset from all 5 domains and train HyperDAS for 5 epochs. We adjuste the learning rate from $2 \times 10^{-5}$ to $5 \times 10^{-4}$ and schedule the sparsity weight $\lambda$ ranging from $0.75$ to $1.5$ starting after $50\%$ of the total steps. This set-up allows the model to first find a stable solution across all domains with soft intervention before forcing it to converge to a single token selection.

**Result** We report the performance of HyperDAS trained over all entity type split in Table 3a. The model performs better than MDAS but **slightly worse** than HyperDAS trained on individual entity type split by **4.0%**. Specifically, HyperDAS-All-Domain performs worse over **city** and **nobel laureate** split, better over **occupation** split, and on-par over **physical object** and **verb** split.

### A.2 DATASET SPECIFICATION

| Domain/Attribute | # of Cause Example | # of IsolateExample | # of Entity |
|---|---|---|---|
| **City** | 34899/7016 | 49500/9930 | 3552/3374 |
| Country | 7925/1544 | 8250/1655 | 3528/2411 |
| Language | 6207/1252 | 8250/1655 | 3471/2221 |
| Continent | 8254/1658 | 8250/1655 | 3543/2567 |
| Timezone | 5371/1144 | 8250/1655 | 3414/1900 |
| Latitude | 3813/743 | 8250/1655 | 3107/1519 |
| Longitude | 3329/675 | 8250/1655 | 2989/1357 |
| **Nobel Laureate** | 39771/6754 | 44628/7600 | 928/928 |
| Country of Birth | 7218/1356 | 8908/1520 | 928/909 |
| Award Year | 11037/1904 | 8930/1520 | 928/926 |
| Gender | 854/96 | 8930/1520 | 592/149 |
| Field | 9518/1558 | 8930/1520 | 928/922 |
| Birth Year | 11144/1840 | 8930/1520 | 928/927 |
| **Occupation** | 54444/1582 | 29052/864 | 799/785 |
| Work Location | 24216/724 | 9684/288 | 799/708 |
| Duty | 12090/371 | 9684/288 | 785/522 |
| Industry | 18138/487 | 9684/288 | 799/600 |
| **Physical Object** | 49114/4659 | 35285/3636 | 563/563 |
| Color | 14707/1518 | 8825/909 | 563/563 |
| Category | 13540/1273 | 8820/909 | 563/562 |
| Texture | 14666/1265 | 8821/909 | 563/561 |
| Size | 6201/603 | 8819/909 | 563/528 |
| **Verb** | 70003/3806 | 14396/782 | 986/984 |
| Past Tense | 34043/1848 | 7188/391 | 986/975 |
| Singular | 35960/1958 | 7208/391 | 986/978 |

Table 1: The details of the dataset used for the experiment, in the format of train/test splits. For every model in each setting. Methods are trained on the full dataset of that setting with 5 epochs. The prompts used by the train/test splits are completely disjoint.

### A.3    DATASET PREPROCESSING

HyperDAS uses an attention mechanism to gather information from the hidden states of the target model $\mathcal{M}$ when running the base and counterfactual sentences. This makes HyperDAS overtly powerful in some situations. Consider the following input:

$$
\begin{cases}
\text{Base} & \bar{\mathbf{x}} = \text{Vienna, known for its Imperial palaces, is a city in the country of} \\
\text{Counterfactual} & \hat{\mathbf{x}} = \text{I love Paris} \\
\text{Instruction} & \mathbf{x} = \text{Localize the latitude of the city}
\end{cases}
\tag{16}
$$

If the model works as intended, it will intervene on the 'Latitude' subspace, which will leave the 'Country' features intact and therefore the target model will predict Austria.

However, since the model can access the hidden states $\mathcal{M}(\bar{\mathbf{x}})$, it knows that the queried attribute in the sentence is 'Country', which is different than the targeted attribute 'Latitude'. Through training, HyperDAS learns a shortcut to a trivial solution—performing no intervention when the target attribute is different from the one mentioned in the sentence. With this shortcut, the **Isolate** objective no longer works and the HyperDAS fails to learn disentangled feature subspaces for different attributes.

Figure 9 shows how the HyperDAS may learn a trivial solution to the RAVEL benchmark if the relevant information (base prompt attribute) can be accessed by the model.

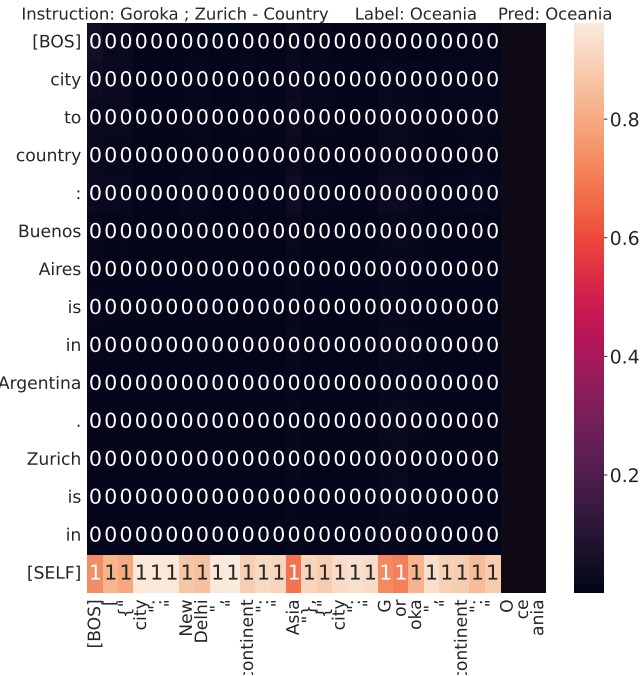

Figure 9: The trivial solution learnt by the HyperDAS on isolate examples when no mask is applied on the attribute token in the prompt. HyperDAS learns to not intervene if it sees the base prompt attribute to be different than the attribute in the instruction.

Therefore, for each pair of prompts $\bar{\mathbf{x}}, \hat{\mathbf{x}}$ at training, we apply an intervention mask to all the tokens starting from the attribute mention. The hidden states from token with intervention mask is not visible to HyperDAS and therefore cannot be selected for intervention. The example becomes:

$$\begin{cases} \text{Base} & \bar{\mathbf{x}} = \text{Vienna, known for its Imperial palaces, is a city in the } \textcolor{red}{\text{country of}} \\ \text{Counterfactual} & \hat{\mathbf{x}} = \text{I love Paris} \\ \text{Instruction} & \mathbf{x} = \text{Localize the latitude of the city} \end{cases} \quad (17)$$

where the hidden states of the red text is masked from the HyperDAS .

### A.4 LOADING HYPERDAS WITH PRE-TRAINED PARAMETERS

We have also explored initializing the HyperDAS from a pretrained LM instead of initializing it from scratch. With Llama3-8b (Meta, 2024) as the target LM, we initialize the modules of HyperDAS , besides the multi-head cross-attention heads and pairwise token position scores attention heads, as the copy of the parameters from the target model. We then evaluate the performance of this variation of the model on the city dataset of RAVEL (Huang et al., 2024).

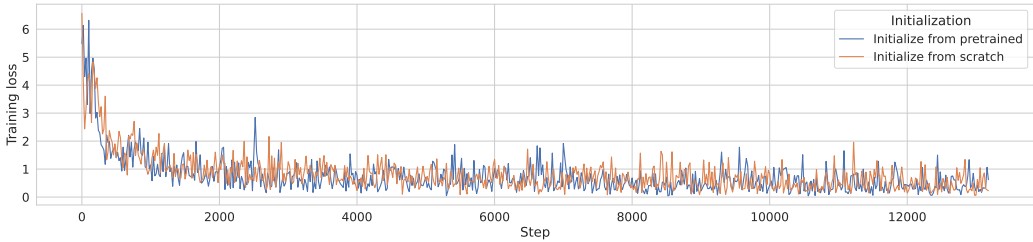

Figure 10: The loss curve of HyperDAS initialized from scratch or from pretrained LM while training on the city dataset of RAVEL.

In Figure 10, we observe that there is no significant difference between the model initialized from scratch and the model initialized from Llama3-8b parameters. However, it remains unknown how would this difference change as the training of HyperDAS scales.

### A.5 SPARSE AUTOENCODERS

We experiment with different feature subspace dimension, as shown in Figure 11. We add an trained **sparse autoencoder** as another baseline. Following the exact same setting in (Huang et al., 2024), we train sparse autoencoder that projects the target hidden states into a higher-dimensional sparse feature space and then reconstruct the original hidden states.

### A.6 ABLATION RESULTS

See ablations in Table 12.

### A.7 INTERVENTION PATTERNS

Here we include a few demonstrations of the intervention pattern that HyperDAS generates on RAVEL, as shown in Figure 13.

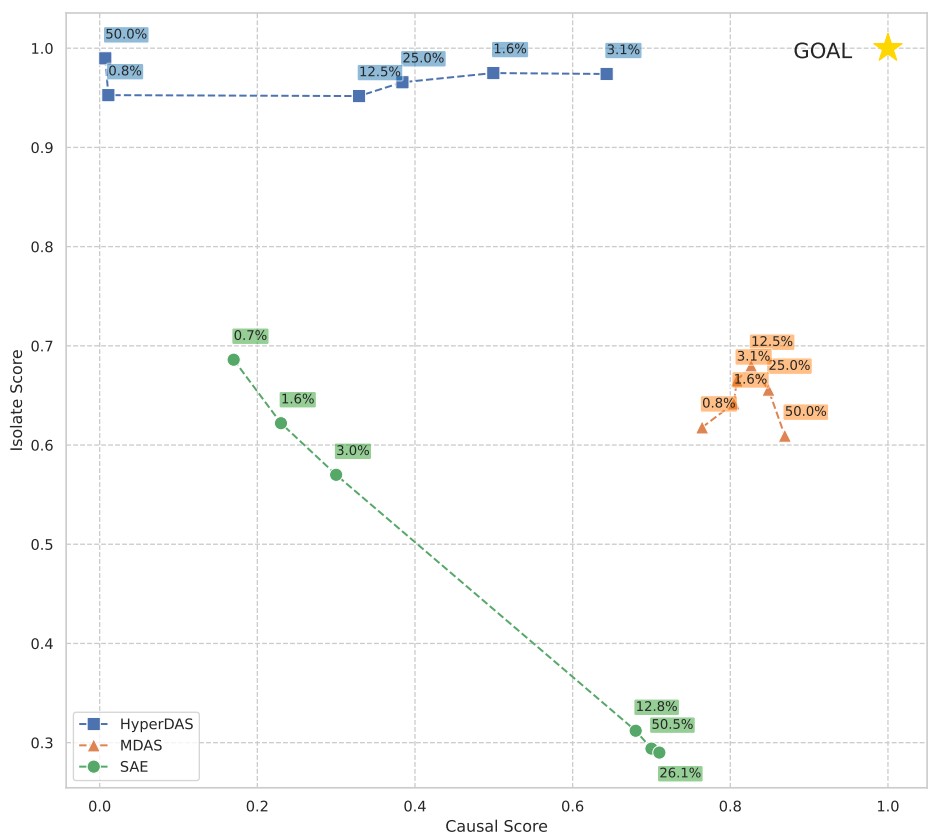

Figure 11: Cause (x-axis) and Iso (y-axis) scores trade-off for HyperDAS, MDAS, and SAE when using different feature size shown as the ratio %. GOAL $(1, 1)$ indicates the score with which the method is able to disentangle the feature subspace perfectly.

| Ablation | Causal | Iso | Disentangle |
|---|---|---|---|
| HyperDAS | 70.8 | 93.9 | 82.4 |
| -No Cross Attention | 68.2 | 83.9 | 76.1 |
| -No DAS | 0.8 | 97.4 | 49.1 |
| -No Hypernetwork | 15.1 | 46.9 | 31.0 |

Figure 12: Ablation results for HyperDAS. **No DAS** has no rotation matrix and intervenes on entire hidden representations. **No Hypernetwork** replaces concept encoding via transformer with a vector lookup. **No Cross Attention** removes attention head submodules connecting the hypernetwork and target model.

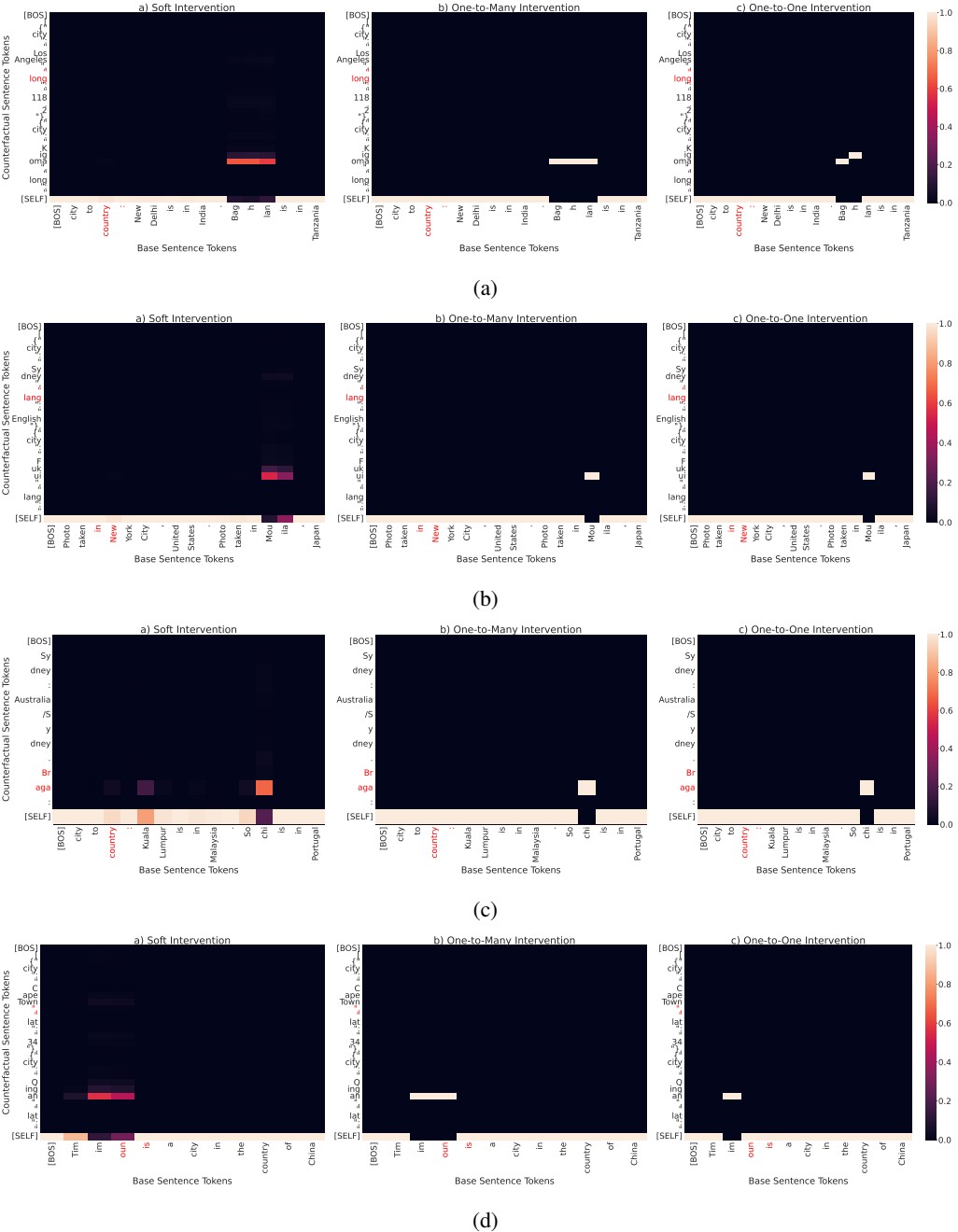

Figure 13: Four types of intervention patterns.

