# OpenReview forum: "HyperDAS: Towards Automating Mechanistic Interpretability with Hypernetworks"
_ICLR.cc/2025/Conference — ICLR 2025 Poster_

### Official Review · Reviewer_Amxz · 2024-10-30

**Soundness:** 4
**Presentation:** 3
**Contribution:** 3
**Rating:** 8
**Confidence:** 3

**Summary:**

This work presents an automated and scalable method for locating features associated with semantic concepts in language models. Given a natural language description of a concept, as well as base and counterfactual prompts, a transformer encodes the description and uses the prompts to locate token positions relevant to the concept, and the internal feature responsible for representing the concept. They evaluate their method on the RAVEL dataset, achieving SOTA performance, and they verify that their method reveals true causal structure via steering experiments.

**Strengths:**

Interesting and creative application of transformers towards automated, scalable interpretability methods

Locating internal semantic concepts has important implications for steering/controlling model behaviour to better align with intentions

**Weaknesses:**

The RAVEL dataset and the notion of localizing features could be explained more clearly

An explanation of the role of the Householder transformation would be useful

**Questions:**

Is there some way of utilizing a pre-trained language model to assist in interpreting the natural language instruction, as opposed to relying solely on a model trained from scratch on RAVEL?

What was the motivation for using the Householder transformation?

How does this approach to locating concepts relate to the approach of representation reading (as in Zou et al. (2023) "Representation Engineering")?

---

> ### Author Response · Authors · 2024-11-16
> **To Reviewer Amxz**
>
> Thank you for your insightful feedback and comments on the paper. Here is our response corresponding to each point you mentioned in your review. A revised paper will be uploaded to address all of the following discussion.
>
> ---
>
> ### Weakness 1: RAVEL and notation
> **Response:** Thank you for your suggestion! We have revised the notations and descriptions in Section 2 and Section 3 for a clearer and more consistent illustration of the RAVEL benchmark.
>
> ---
> ### Weakness 2: An explanation of the role of the Householder transformation
> **Response:** In general response (1), we provide a detailed explanation of the design choice to train a low-rank orthogonal matrix and a downstream projection for a Householder vector. We have revised the paper to make it clearer.
>
> ---
> ### Question 1: Does initializing from pre-training help?
> **Response:** This is an interesting suggestion. Intuitively, loading the HyperDAS from pre-trained parameter would not help since:
> The extra cross-attention block reads from and writes to completely different distribution of the hidden states
> The residual stream integrates the information from the base/counterfactual hidden states via cross-attention, making the input of the MLP and self-attention module to be different.
>
> To verify this intuition, we have trained HyperDAS from scratch / with pre-trained parameter on RAVEL-city and reports its disentangle score over steps (See Appendix A of the revised paper). We confirm that **no significant advantage** is observed for the HyperDAS trained from the pre-trained parameter. We do acknowledge that this could be different if we scale up the training.
>
> ---
> ### Question 2: What was the motivation using Householder transformation
> **Response:** Similar to W2, please checkout general response (1)

---

> > ### Comment · Reviewer_Amxz · 2024-11-25
> >
> > I thank the reviewers for addressing my concerns. I will maintain my score

---

### Official Review · Reviewer_iDST · 2024-11-04

**Soundness:** 3
**Presentation:** 2
**Contribution:** 2
**Rating:** 5
**Confidence:** 4

**Summary:**

The authors present a method for identifying directions in the activation space (hidden state) of language models corresponding to chosen concepts of interest. In particular they present a technique which takes as input a base prompt, a counterfactual prompt and a concept label. It then attempts to isolate and transfer the representation of the concept from the hidden state of the counterfactual prompt to the hidden state of base prompt (by intervening on the the forward pass of the base prompt).

Their technique is a new variation of the DAS method. DAS learns a rotation and projection of the the activation space that isolates a causally mediating direction for a particular concept.
Their new method, HyperDAS, uses an additional transformer hypernetwork to choose which token positions in the base prompt to intervene on and which token positions in the counterfactual prompt to read activation values from. The hypernetwork uses attention blocks which read keys and values from the base and counterfactual hidden states.

**Strengths:**

Their method seems like a necessary and logical extension to address the problem of token matching when performing interventions with the DAS method. They report state of the art results on the concept disentanglement benchmark. This line of research seems like a promising approach to better understand the internal states of neural networks.

The authors conduct detailed ablation experiments that help to isolate which aspects of their methodology are most important to achieve strong performance.

They also are careful to consider whether their intervention is philosophically justified, or whether they are adding too much complexity to faithfully interpret the model. I feel mostly persuaded that their method does in fact uncover properties of the underlying model, rather than learning new representations (or at very least, it does not seem to be worse in this regard than DAS).

**Weaknesses:**

Figure 3(b) makes the claim that HyperDAS beats MDAS appear less impressive. At many intervention layers, MDAS appears to be superior, so this result feels somewhat cherry-picked, although it is true that layer 15 is by a small margin the best layer for both methods. (However, the claim on line 377 seems to contradict my reading of the graph, so I may be misunderstanding something).

Although it is briefly explored in one of the ablations, the authors do not adequately explain why they train the hypernetwork to take natural language input. They train a separate hypernetwork for each domain, so it seems that there would be a limited number of possible queries, and they would be no need to train a general language model such that the intervention can be specified in natural language.

Nitpicks:
- Abstract: “identifying neural networks features” -> “identifying neural network features”
- Line 142: “targe concept”
- Line 159: the index j is not defined (and clashes with the j-th token index used later)
- Line 481: “For examples”

**Questions:**

- Is the hypernetwork highly specialized to the distribution of the RAVEL benchmark, or can it be used to isolate concepts using real texts from more natural sources?
- Can a single network be trained that performs all of the tasks in RAVEL? It seems it should be possible to train a hypernetwork to make very general interventions, given the natural language input.

---

> ### Author Response · Authors · 2024-11-16
> **To Reviewer iDST**
>
> Thank you for your insightful feedback and comments on the paper. Here is our response corresponding to each point you mentioned in your review. A revised paper will be uploaded to address all of the following discussion.
>
> ---
> ### Weakness 1: Layer-wise result between MDAS and HyperDAS
> **Response:** This is a valid concern that should be resolved by new experimental results included in the updated paper.
>
> We chose layer 15 based on the fact that the original RAVEL paper had success on layers in the middle. Due to computational constraints, we had only run HyperDAS and MDAS on the subset of the city domain where the country attributes from other attributes. The resulting figure that we included in the initial submission was misleading and you are right to point out that these results looked cherry picked! The results showed that MDAS was better than HyperDAS at several layers.
>
> We now have the results of disentangling all pairs of attributes. It shows that MDAS disentangles ‘country’ from other attributes well, but HyperDAS is able to succeed on all attributes, which is the actual task for RAVEL.
>
> We have updated the figure in the new version of the paper.
>
> ---
> ### Weakness 2: Training HyperDAS over all domains
> **Response:** We have provided our detailed rationale and plans for future work regarding this insight in general response (3). We totally agree that this is an important next step for the work.We have provided our detailed rationale and plans for future work regarding this insight in general response (3). We totally agree that this is an important next step for the work.
>
> ---
> ### Weakness 3: Typos
> **Response:** Thank you for such a thorough review on the paper. We have revised the paper accordingly to all the mistakes you have spotted. We have defined and clarified all the subscripts and superscripts in the paper to have fixed and distinct meanings.
>
> ---
> ### Question 1: Is HyperDAS specific to RAVEL?
> **Response:** Thank you for this question! Please see the general response (3)
>
> ---
> ### Question 2: Can a single HyperDAS be trained across all domains
> **Response:** Thank you for this question! Please see the general response (3)

---

> > ### Comment · Reviewer_iDST · 2024-11-17
> >
> > Thanks for your update regarding the figure. The new results look much stronger.
> >
> > Although this is somewhat addressed by your general answers, it would be helpful if you could provide a very explicit answer to my question above before I come to a final conclusion.
> >
> > > Although it is briefly explored in one of the ablations, the authors do not adequately explain why they train the hypernetwork to take natural language input. They train a separate hypernetwork for each domain, so it seems that there would be a limited number of possible queries, and they would be no need to train a general language model such that the intervention can be specified in natural language.
> >
> > My understanding based on your general response (3) is that in the current experiments, the natural language instruction to the hypernetwork is fixed (or possibly takes a small number of different inputs?) for any given network. So the hypernetwork does not in any meaningful sense "understand" the instruction, and this input is not really doing much. So while it is an exciting direction for future work, there is not currently any solid experimental evidence that this can actually be made into a generic natural language interface for querying the internal representations of a model.

---

> > > ### Author Response · Authors · 2024-11-25
> > > **Re: Official Comment by Reviewer iDST**
> > >
> > > Your understanding of general response (3) is essentially correct. The hypernetwork currently takes in a natural language input that specifies the entity type and the attribute targeted. There are 23 valid entity-attribute pairs in the RAVEL dataset across five splits of entities, namely, City, Nobel Laureate, Verb, Physical Object, and Occupation. This means there are 23 possible inputs to the HyperDAS models we train. Our most recent general update reports that we can train a HyperDAS model on all 23 possible inputs that beats the previous state-of-the-art method of MDAS.
> > >
> > > Please let us know if we can clarify the situation further!

---

> ### Comment · Reviewer_iDST · 2024-11-25
>
> Thanks, this is clarified. I will stick with a score of 5, as I feel this limitation to the instruction is fairly central to the value of the paper's contribution. Even when all 23 possible inputs are trained on a single network, this is a far cry from a general natural language instruction.

---

### Official Review · Reviewer_7H7A · 2024-11-04

**Soundness:** 3
**Presentation:** 3
**Contribution:** 3
**Rating:** 6
**Confidence:** 5

**Summary:**

This work proposes a hyper-network based approach to enforcing causal interventions  in foundation models. Given an instruction and two text inputs, one base input and a counterfactual input, the model aims to localize the token in the base input which answers the instruction and instead return the counterfactual input answer. All other aspects of the output should remain unchanged. To achieve this the model has two output heads: one for finding the corresponding answers in the two inputs, and one for transforming the token embedding of the base entity to the counterfactual embedding for the corresponding entity. Some ablations are conducted on the localization head which requires sparsity to work effectively. It is shown that for most entities in the RAVEL dataset the HyperDAS approach outperforms the MDAS baseline, especially at the Iso score which depicts that interventions are more controlled than for MDAS. This increase in Iso score never corresponds to a drop in Causal score and so the overall disentanglement score of the model is improved over MDAS.

**Strengths:**

# Originality
The use of a hyper-network to learn to automate causal interventions is novel to my knowledge. The exact implementation itself also appears novel, with the location and intervention heads being a modular and measurable approach to training the interventions. This also means that the separation into Causal and Iso scores is possible and this is useful for evaluating the model.

# Quality
Overall the model work is of a high quality with clear hypotheses and appropriate experimental design. I find the separate consideration of Iso and Causal scores to be useful and lend insight into the behaviour of the model. The more detailed consideration of the localization head is also useful and are important ablations in the study. In most cases limitations are clearly acknowledges - for example on lines 421 to 429 where it is mentioned that the model is sensitive to the sparsity hyper-parameter.

# Clarity
Overall the  paper is well written and language appropriate. The structure of the work is also intuitive and aids with the understandability of the work. Figures are clear and legible with helpful captions.

# Significance
The improvement at disentanglement of the model over the baseline presents a clear step forward. In addition the detailed ablations and insight into the localization head could lead to future work. Overall I think this work is of clear interest to the field and makes a clear contribution towards causal interventions in transformer models and working with foundation models.

**Weaknesses:**

# Clarity
Some portions of the model are not fully explain. For example in Equation 1 the inverse of the low-rank orthogonal matrix R is used ($R^{-1}$) but if this matrix is low rank how is it inverted? Is this a pseudo-inverse? Another example is how the localized token positions are used to apply an intervention. In Figure 1 (left) it is shown that the positions are fed into the intervention module but Figure 1 (right) does not show this. I think exactly what the input and output of each head is could be described in far more detail. For an intricate model this is very important and limits clarity. Another part of the model which is not explained is the learning of the rotation matrix. Firstly, why is it necessary to learn $R'$ first and then apply the Householder transformation? Why not directly learn $R$? What does the Householder represent in an embedding space such that it performs a useful computation here? I do think on the whole the model and its workings become evident but this could still be far clearer and more explicitly explained.

# Quality
On the side of quality I think a couple limitations are still not given due consideration. Particularly, the fact that a hyper-network is trained for every entity type. This seems to add a huge amount of computational overhead and limits the scalability of this approach. However, beyond line 92 I do not see this being discussed anywhere. Similarly, there are a couple statements such as the claim that the method helps ``crack open black box models'' on line 415 which do not quite seem true. I am not certain how this  approach does this and it is not clearly explained. Similarly I think there is a lingering assumption that a single token in the base input maps to a single counterfactual token. This seems to be a property of the dataset (which is totally fine) but then I am not certain the black box has been cracked open when the model then identifies this property of the dataset. Lines 514 and 515 have a similar issue for me. Perhaps I am not appreciating some nuance to this, but I think the model makes a clear contribution without needing to be too far reaching in its claims. For example, the ability to easily and clearly manipulate the black box seems equally impressive to me - especially when considering the precision of the approach demonstrated by the Iso score. Similarly, for Figure 6 - it is stated that HyperDAS learns different feature subspaces for different attributes but in general the clusters are very tightly packed. I don't think that this statement is clear from the figure but also doesn't seem to be the main point of the work anyway. Lastly, I think more consideration should be given to the fact that when using too much sparsity loss the model does not behave appropriately but still obtains a near perfect disentanglement score. This demonstrates a limitation of the experimental design. I note that this phenomenon is  noted in the work, but it stops short of actually acknowledging it as limitation of the method which is necessary. I recommend the work is revised such that the claims are made more exact.

If I am not misunderstanding something, then I would increase my score one level for each of the two main points above: 1) more clarity needs to be given on the exact details of the model, 2) the claims needs to be made more precise.

**Questions:**

I have left some questions in the weaknesses section above which I would like answered. However two clear question are:

- How much extra computational overhead is added when using HyperDAS over MDAS?
- A small one: is $y$ missing from the end of Equation 13?

---

> ### Author Response · Authors · 2024-11-16
> **To Reviewer 7H7A**
>
> Thank you for your insightful feedback and comments on the paper. Here is our response corresponding to each point you mentioned in your review. A revised paper will be uploaded to address all of the following discussion.
>
> ---
> ### Clarity 1: How to get the inverse of low-rank orthogonal matrix R?
> **Response:** Given a low-rank orthogonal matrix $R \in \mathbb{R}^{n \times m}$, by definition its inverse $R^{-1} \in \mathbb{R}^{m \times n}$ is its transpose $R^{-1} = R^T$
>
> ---
> ### Clarity 2: How the localized token positions are used to apply an intervention?
> **Response:** We have revised Fig 1, Sec 2 and Sec 3 to make the process as clear as possible. In a few sentences, the localized token positions, which is a matrix of “intervention weight” on the token pairs. Each column of the matrix is a distribution sums up to 1 and represents the portion of the counterfactual token to intervene on this base token. At training time the source hidden states is a weighted sum given the distribution (Equation 10 in Sec 3). This weight is snapped to be 0 or 1 and a one-to-one token alignment is enforced during test time evaluation.
>
> Given a source hidden states and a base hidden states, we extract their features in the concept subspace respectively, and perform the interchange intervention (Equation 11 in Sec 3).
>
> ---
> ### Clarity 3: Why is it necessary to learn R’?
> **Response:** In general response (1), we provide a detailed explanation of the design choice to train a low-rank orthogonal matrix and a downstream projection for a Householder vector. We have revised the paper to make it clearer.
>
> ---
> ### Clarity 4: What does the Householder represent in an embedding space?
> **Response:** The Householder vector represents a reflection operation of the base rotational matrix with respect to the given vector, which is different based on the input sentence and the target concept in the instruction.
>
> ---
> ### Clarity 5: General clarity of the model details
> **Response:** We have revised the entire Section 2 and Section 3 systematically to make the detail of the model as clear as possible. Any new feedback and suggestions on top of that is highly appreciated.
>
> ---
> ### Quality 1: Training HyperDAS over all domains
> **Response:** We have provided our detailed rationale and plans for future work in general response (3).
>
> ---
> ### Quality 2: Overclaiming of ‘cracking open black box model’ and understanding and interpreting the internal workings of complex language models.’
> **Response:** We understand the reviewers concern, and we have removed the line 415 about ‘cracking open the black box model’ and modified the line 515 in the conclusion to state that we are optimistic, but haven’t shown this conclusion definitively. While we believe that our work is contributing to the understanding of black box models, we hope that this brings our language more in line with the results presented!
>
> ---
> ### Quality 3: Assumption of one-to-one token correspondence
> **Response:**  The previous state-of-the-art on the RAVEL benchmark was MDAS which aligned only one token in the base to one token in the counterfactual. We ran some tests and found that 47% of the time HyperDAS selects only one token, so the model does take advantage of this capability. We think this is an interesting point that should be highlighted, so we have included a new discussion point on it in the main test.
>
> ---
> ### Quality 4: Why does strong sparsity loss demolish the model performance
> **Response:** In general response (2), we provide a detailed explanation why we feel principled decisions were made to avoid the issues that come with soft interventions and sparsity.
>
> ---
> ### Question 1: Computation Overhead
> **Response:** Thank you for pointing out this key comparison. HyperDAS enables searching for a better localization of the concept, and therefore is naturally more computationally expensive. To capture how much, we trained both HyperDAS and MDAS over the RAVEL-city domain and reported the cost, training speed, and convergence speed (See Appendix A of the revised paper). Our HyperDAS model has 10x memory cost compared to MDAS, i.e. more parameters are loaded in, to **reach the same training speed per token**.
>
> ---
> ### Question 2: Missing Y in Equation 13?
> **Response:** Yes, the label y was missing in the equation of the cross-entropy loss. We have fixed it in the revised version.

---

> ### Author Response · Authors · 2024-11-25
> **To Reviewer 7H7A - Follow up**
>
> Thank you for the suggestion. We've trained a single HyperDAS network across all five domains, achieving better performance than the baseline method of MDAS. Our experiments demonstrate that HyperDAS can be effectively trained for causal intervention across multiple domains, providing better evidence of scalability.
>
> For further details, please refer to the Update in the general response and Appendix A.1 of the paper.
>
> Additionally, we have also revised the Figure 6 & 7 by repeating the exact same experiments on subspace clustering and similarity but raise the number of random examples picked from 1k to 100,000. Now the figures correctly reflect our statement in the discussion.
>
> Please let us know if we can clarify the situation further!

---

> > ### Comment · Reviewer_7H7A · 2024-11-25
> > **Response to Authors**
> >
> > I thank the authors for their response to my comments. I have noted the general comments as well and the revised paper.
> >
> > As it stands I do not think the changes go far enough to address my concerns and as far as the computation cost goes, I am more concerned. A 10x memory cost, to me, is not something to be put in the appendix. This is a significant cost to achieve the performance improvements noted. This should be discussed. I do not think a higher memory cost should be grounds for rejection - indeed as I pointed out in my review I think the results are interesting already. I do think it is grounds for rejection to gloss over such an important point. Training speed per token is one perspective on compute costs too and I think more discussion and metric in general are needed for this to be done justice. To quote the appendix which I think puts this more into perspective: MDAS costs 6.4G of memory. HyperDAS costs 68G.  Unfortunately, if this is not corrected in the main text I will lower my score.
> >
> > For clarity, I have looked through the revised paper at it seems there are not many changes. For example, the paper still has $R^{-1}$ in the equations where $R^T$ is used. This is a minor point but a clear example of how implementation details are skipped over. The Householder transformation (including the discussion in the rebuttals) is still unhelpful. I know what a Householder transformation is - what is missing is why you chose to use it and the effect on the embedding space. If I said that I performed clustering on an embedding space I would be able to say that this would pick out words with semantically similar meaning. I would like a similar point for this transformation justifying the computation.
> >
> > A new question from the general comment: If R is enforced to be orthogonal using torch.orthogonal, does this mean there is no mathematical restriction enforcing the orthogonality? Is any information lost by taking this approach?
> >
> > Thus, I will wait to consider the final revision to the paper before accepting. However, as it stands I still have concerns on the clarity and missing implementation details.

---

> > > ### Author Response · Authors · 2024-11-26
> > >
> > > ## We made a real effort to address your concerns on clarity and presentation
> > >
> > > First, we must reject the characterization that there “are not many changes”. A simple comparison between the current PDF and the initial submission side-by-side will show that the material from Section 3 describing the HyperDAS architecture was greatly expanded on.
> > >
> > > 1. The material with the paragraph header ‘Cross-attention Decoder Layer’ doubled in size with more prose and an additional equation.
> > >
> > > 2. The material with the paragraph header  ‘Pairwise Token Position Attention’ was rewritten entirely with greatly simplified notation and additional prose.
> > >
> > > 3. The material with the paragraph header  ‘ Feature Subspace Rotation Matrix’ also doubled in size with additional prose to explain why we use the householder vector.
> > >
> > > 4. The material with the paragraph header  ‘Interchange Intervention’ tripled in size with more prose explanation.
> > >
> > > We understand that you might not have found these changes helpful, but we took the time to carve out enough room in this paper to expand on this section in an attempt to address your concerns. We think the paper improved greatly from this process!
> > >
> > > ## Computational Overhead
> > > We have conducted a more thorough analysis of computational overhead that includes FLOPs. We also properly account for the memory usage of the target Llama model, which we did not do previously.
> > >
> > > *HyperDAS is more powerful than MDAS, but also more computationally expensive. Training our HyperDAS model for one epoch on disentangling the country attribute in the city domain takes 468923 TeraFLOPs while training an MDAS model for one epoch on the same task takes 193833. HyperDAS requires roughly 2.4x compute. Our target Llama model requires 16GB of RAM while the HyperDAS model requires 52GB more and MDAS requires 4.1GB more per attribute. The memory usage of HyperDAS does not go up with additional attributes, so when trained on all of RAVEL together (23 attributes), MDAS (23*4.1 + 16 = 110.3GB) would exceed the memory usage of HyperDAS (52 + 16 = 68GB).*
> > >
> > >
> > > The above prose is now the second discussion point in our main text! To be clear, it was never our intention to gloss over these details; we are already at page limit and working in additional material to the main text is difficult.
> > >
> > > We also have new results showing that we can achieve an 81.7 overall disentanglement score on RAVEL using 2 transformer layers instead of 8 for the hyper network. This brings the compute to 415711 TeraFLOPs for HyperDAS which is 2.14x what MDAS requires. We will continue to experiment in an attempt to push down this number.
> > >
> > >
> > > ## Householder Transformation and the Rotation Matrix
> > > We choose to use the householder transformation because we needed a way to use (1) a vector that embeds the target concept and manipulate (2) the static orthogonal matrix R that targets a fixed subspace in order to produce (3) a new orthogonal matrix R' that targets a new subspace that contains the target concept. The householder was a linear algebra operation that satisfied these criteria. **We have rewritten the prose in the main text on householder transformations to explain this design process better.**
> > >
> > > We have now replaced all instances of the inverse operation with the transpose operation in the text for consistency. Because R has orthogonal columns, the transpose of R is the left inverse of R. This is the reason that we used inverse and transpose interchangeably when writing about the matrix R, which is enforced to be orthogonal with torch.orthogonal.

---

> > > > ### Comment · Reviewer_7H7A · 2024-12-03
> > > > **Response to Authors**
> > > >
> > > > I appreciate the effort put in by the authors to accommodate my suggestions. I appreciate that space is limited, and given this fact I think the changes are sufficient. Having read the final version of the paper (for review) I am satisfied that the clarity is greatly improved from the original draft. I also appreciate that the reviewers took my concern on the compute seriously and have now included a more thorough and clear discussion on this topic. Overall I am now satisfied that this work is of an appropriate standard for acceptance and I am confident on this. I would urge the authors to continue to work on clarity for a published draft. Thus, I am raising my score to 6 and confidence to 5.

---

### Official Review · Reviewer_vVg5 · 2024-11-06

**Soundness:** 3
**Presentation:** 3
**Contribution:** 3
**Rating:** 8
**Confidence:** 3

**Summary:**

This work proposes a method for automating the selection of particular linear directions in feature space that represent interpretable concepts or features. In prior work, model steering or activation patching has been performed utilizing optimization or datasets of prompts, but each method requires some manual effort or search in order to pair a particular neuron to a corresponding concept. This work proposes to use a hypernetwork that is conditioned on a counterfactual prompt, as well as intermediate features of an LLM with respect to a base prompt. The hypernetwork predicts for each token of the counterfactual prompt, its corresponding position in the base prompt in addition to an aligned counterfactual representation that is able to "override" the base prompt features.

The hypernetwork is trained on RAVEL, with cross entropy to enforce correct token pairings, as well as a sparsity loss to encourage a one to one mapping between tokens. When evaluating on RAVEL, generations are scored according to whether or not the target attribute was successfully changed, and also on whether or not untargeted attributes were left alone. Evaluations compared to MDAS are favorable, and ablations show

----------------------------------------------------------------------
I believe this work will be of interest to the community. I hope that the discussed changes and additions make it into the final camera-ready. I will keep my score.

**Strengths:**

Originality / Significance:
An automated method for finding / aligning concept directions at the token level is significant. I also think its important to note that it even performs better in evals than non automated methods.

Clarity/Writing: I found the paper to be well written.

Results:
One thing I found particularly compelling was the ISO score of HyperDAS on RAVEL. When performing interventions for model steering (or even SAEs), there is often a failure to isolate/disentangle particular directions, yielding steering in a spurious direction. A very high ISO here is a good sign that the discriminative power of the hypernetwork is high.

I also appreciated the discussion of what we're really investigating or uncovering when we train supervised interpretability tools.

**Weaknesses:**

The largest weakness of this work is lack of baselines or evaluations, there is the one dataset and one baseline method. Not that interpretability methods have to be quantitatively better than others, but contextualization helps. I would encourage showing additional baselines.

This does not influence my score, but I am broadly concerned that any method trained to perform these interventions is adding information that is not in the model under investigation.

**Questions:**

1. What makes this a hypernetwork? Hypernetworks should predict some weights of a target network.

2. Since you don't actually clamp features to 0, sparsity is only softly encouraged with the sparsity loss, meaning there is always some feature entanglement. I'm confused about why adding too much sparsity loss results in poor performance. Doesn't better disentanglement imply that we're targeting only the target attribute?
More specifically, figure 8 states "[...] it demolishes the model’s ability to form interpretable intervention patterns and adhere to specified constraints". What does this mean?

---

> ### Author Response · Authors · 2024-11-16
> **To Reviewer vVg5**
>
> Thank you for your insightful feedback and comments on the paper. Here is our response corresponding to each point you mentioned in your review. A revised paper will be uploaded to address all of the following discussion.
>
> ### Weakness 1: Why only use the RAVEL benchmark?
> **Response:** The task of disentangling information in the residual stream of a transformer tests a model’s ability to identify the correct subspace for a concept. Without the pressure of needing to **cause** a concept to change while **isolating** this concept from others is what creates a challenging task for the HyperDAS model. We want the localization to be as precise as possible, meaning the discovered subspace/activations corresponding to the concept should be as disentangled from the other concepts as possible. RAVEL was built exactly for this purpose.
>
> However, we understand the desire for more datasets and will run experiments on the function vector dataset from Todd et al., 2024, in which the authors discover a set of attention heads activating on an in-context learning prompt could causally trigger the model to perform the task. We plan to run the experiment on recovering the discovery made in the paper automatically with HyperDAS. We will report the result if it could be finished within the discussion period. Crucially, this won’t challenge the ability of HyperDAS to find the correct subspace, because there is no **isolation** objective that punishes solutions that intervene on other concepts as well.
>
> ### Weakness 2: Why only use MDAS for baseline?
> **Response:** We acknowledge the importance of including significant baselines to holistically compare the method with other interpretability methods. We only reported the result of MDAS (Huang et al., 2024) as it was indisputably the best method on the RAVEL benchmark compared with 7 baseline methods, including hidden states patching and sparse autoencoder.
>
> Nevertheless, the original paper used a different LM (Llama-2) as the target model. To ensure that HyperDAS perform better than the baseline methods, we reproduced the experiments on RAVEL-city with SAE and reported the result in Appendix A. Subsequently, we will revise the paper again to include the SAE and other baselines (PCA, differential binary mask, etc.) in the main results table.
>
> ### Question 1: What makes this a “Hypernetwork”?
> **Response:** Yes, we agree that the term “hypernetwork” is often used specifically to refer to processes that update weights. We have adopted a broader sense for the term that encompasses situations in which one network manipulates the representations of another. We will clarify this terminological shift, and we are open to changing the term if there is a concern that it will confuse people.
>
> ### Question 2: Why does strong sparsity loss demolish the model performance?
> **Response:** In general response (2), we provide a detailed explanation of why a strong sparsity loss leads to model overfitting on the soft intervention. We will revise the paper to make the point clearer.

---

> > ### Comment · Reviewer_vVg5 · 2024-11-23
> >
> > I appreciate the detailed response to my comments, as well as the additional results for the SAE. While RAVEL and MDAS may be the best possible fit to evaluate the performance of this method, I think that adding additional results where possible can only strengthen the message.

---

### Author Response · Authors · 2024-11-16
**To All the Reviewers (2)**

## 3) Excitement around scaling HyperDAS (**all reviewers**)

Our results show that we can train HyperDAS to achieve state of the art performance on the RAVEL benchmark by training a separate model for each entity type split, which is the set up used to train the previous state of the art MDAS. In future work, we want to move away entirely from the RAVEL dataset and train a HyperDAS model with a generic natural language interface for localizing concepts to model internals. This HyperDAS could analyze a model on command. As reviewers point out, this might require leveraging pretrained language models for the Hyper Network and building out an expansive dataset that goes beyond the entity-attribute framing of RAVEL, e.g. *localizing the value of variable ‘x’ on line 56* when the prompt is code or localizing *the plan to move a knight when the opponent moves their queen* when the prompt is a chess game. We are excited to build on the foundation established by the current paper  and have high hopes!

---

### Author Response · Authors · 2024-11-16
**To All Reviewers (1)**

Thank you all for your insightful feedback on the paper’s experiments and presentation. We have made the first update of the paper PDF to help the reader understand the method better:

We have cleaned up notations and expressions in Sec 2 and 3 to make it more readable and consistent. We:
- Fixed all the typos mentioned by reviewer iDST
- Add SAE result as a baseline; See Appendix A.4
- Add Section “Computation Overhead” to address the question from reviewer 7H7A; See Appendix A.5
- Add Section “Loading HyperDAS with Pre-trained Parameters” to address the question from reviewer Amxs; See Appendix A.3
- Update Fig 3b to show comparison between HyperDAS and MDAS over the entire city domain, addressing the cherry-picking concern from reviewer iDST. The previous figure mistakenly reported a limited subset of the city domain that was misleading. Many thanks again to iDST for spotting this.

We are actively working on further baseline experiments and revised paper that we will include in a second update.

Here we summarize our response and revision to some important questions that multiple reviewers share:

---

## 1) Clarification on the use of Householder transformation (Reviewer **7H7A, Amxs**)

The rotation matrix $R$ is fixed regardless of which concept is being targeted for intervention and is enforced to be orthogonal using torch.orthogonal. However, we want our hypernetwork to be able to **dynamically select** which linear subspace to intervene on, e.g. intervene on one subspace when targeting the country of a city and intervene on a different subspace when targeting the continent of a city. To enable this, we allow the hypernetwork to perform a householder transformation of R that is conditioned on the concept targeted. This results in a new dynamically constructed orthogonal matrix $R’$.

**Definition:**
A Householder transformation $H$ is a matrix of the form:

\begin{equation}
H = I - 2 \frac{\mathbf{v}\mathbf{v}^T}{\mathbf{v}^T \mathbf{v}}
\end{equation}

where $\mathbf{v}$ is a non-zero vector. This matrix is orthogonal ($H^TH=I$) and symmetric ($H^T=H$), for which the transformation $H(R')=HR'$ reflects the subspace $R'$ from the Householder vector $\mathbf{v}$ and keeps its orthogonality.

The last tokens hidden state at the final layer of the HyperDAS decoders, encodes the concept that will be targeted for intervention the Householder vector v that transforms the rotation matrix $R$.

**Why not directly using Householder matrix as the feature subspace:**

Using orthogonal matrix $M$ to encode feature subspace gives us the benefit to project into and back from the subspace due to the property $M^{-1}=M^T$. Indeed, we could use Householder matrix depending on the representation vector v to encode the subspace since it’s orthogonal. However, it’s a square matrix that has to have a dimension of hidden_dim * hidden_dim to match the hidden representation’s dimension. A full-rank hidden_dim-dimensional matrix is essentially just the hidden space after linear transformation, which will result in a full interchange of the hidden states.

---

## 2) Sparsity Loss and Evaluation Mode (Reviewer **vVg5, 7H7A**)

HyperDAS is designed to automate causal interpretability of LLMs through the selection of intervention location and feature subspace in a differentiable way. To make it differentiable, we adopt soft selection of intervention locations across all base-counterfactual token pairs and incentivize it to converge to a sparse solution. Crucially, **during test time evaluations we snap the masks to binary values so that there is a one-to-one alignment between base and source tokens.**

However, we encountered a problem where HyperDAS will learn to align a single source token with **all** the visible tokens in the base sentence. (See Fig 8 middle) This model is performant during training, however, when we snap the masks during test time evaluation, the model fails completely. (Fig 2 right and Sec 3.3)

Therefore, we have experimented with multiple sparsity losses and chosen the one with the best performance (Equation 14) to disencourage the model to select multiple tokens or a linear combination of them.

Yet, this “sparsity” loss term does not punish the situation where a small portion of the token is selected whose sum is less than one. Therefore, if an exceedingly strong sparsity loss is applied at the beginning of the training, the model would learn to distribute the intervention across all the token pairs (Fig 8 right) to “hack” the causal intervention, which is not interpretable either.

Our test time evaluations use one-to-one alignments between base and counterfactual tokens. As long as this is the case, softening the selection operation during training and using sparsity losses to help the model solve the task should be fair game.

---

### Author Response · Authors · 2024-11-25
**To All Reviewers - Update**

### **We trained a performant HyperDAS on all of RAVEL**

Multiple reviewers expressed concerns about the need to train HyperDAS separately for each type of entity. In response to this concern, we conducted further hyperparameter tuning and found a setting for HyperDAS that is performant when trained on all of RAVEL at once. In particular, HyperDAS trained on all domains achieves a score of **80.7**, which is a bit better than the pervious state-of-the-art MDAS at 76.0, but a bit worse than the HyperDAS trained on each domain separately at 84.7.

A new row “-All Domains” has been added to the main table 3a, which is the performance of HyperDAS trained over all the entity splits. A new Appendix section “HyperDAS Over All Domains” has been added where we describe the hyperparameter choices needed to replicate the result.

We hope this demonstrates the potential for scaling HyperDAS!

---

### Comment · Area_Chair_MVao · 2024-11-25
**Discussions between reviewers and authors**

Time for discussions as author feedback is in. I encourage all the reviewers to reply. You should treat the paper that you're reviewing in the same way as you'd like your submission to be treated :)

---

### Meta-Review · Area_Chair_MVao · 2024-12-20

**Metareview:**

This paper aims at automatically selecting linear directions in feature space for interpretability via a hypernetwork. The work builds on top of the original DAS algorithm, where the hypernetwork is trained to predict the representations of the base and counterfactual inputs using  a text specification of the target concept. Evaluations are conducted on the RAVEL dataset using Llama3-8B.

Reviewers agree that this paper presents an interesting improvement over the original DAS algorithm. The main questions were comparisons to MDAS, computational overhead, etc. The author feedback has largely addressed these concerns.

One concern is raised during AC-reviewer discussion: "Although the paper is a solid extension to the original DAS method, but the instruction model is not really processing a natural language instruction in their experiments." Most reviewers and I agree with this concerns, therefore I would request the authors to clarify on their setting more to avoid hyped statement.

**Additional Comments On Reviewer Discussion:**

Most concerns were addressed. During the AC-reviewer discussions one concern is further raised (see meta review), but overall reviewers are in favour of acceptance.

---

### Decision · Program_Chairs · 2025-01-22

Accept (Poster)